# Sleep and White Matter in Adults with Down Syndrome

**DOI:** 10.3390/brainsci11101322

**Published:** 2021-10-05

**Authors:** Victoria Fleming, Brianna Piro-Gambetti, Austin Bazydlo, Matthew Zammit, Andrew L. Alexander, Bradley T. Christian, Benjamin Handen, David T. Plante, Sigan L. Hartley

**Affiliations:** 1Waisman Center, University of Wisconsin-Madison, Madison, WI 53705, USA; vlfleming@wisc.edu (V.F.); gambetti@wisc.edu (B.P.-G.); mbazydlo@gmail.com (A.B.); mzammit@wisc.edu (M.Z.); andy.alexander@wisc.edu (A.L.A.); bchristian@wisc.edu (B.T.C.); 2School of Human Ecology, University of Wisconsin-Madison, Madison, WI 53706, USA; 3Department of Medical Physics, University of Wisconsin-Madison, WI 53705, USA; 4Department of Psychiatry, University of Wisconsin-Madison, WI 53719, USA; dplante@wisc.edu; 5Department of Psychiatry, University of Pittsburgh, Pittsburgh, PA 15213, USA; bhanden@pitt.edu

**Keywords:** Down syndrome, Alzheimer’s disease, white matter, sleep, diffusion tensor imaging

## Abstract

Adults with Down syndrome are at a high risk for disordered sleep. These sleep problems could have marked effects on aging and Alzheimer’s disease, potentially altering white matter integrity. This study examined the associations between disordered sleep assessed via an actigraph accelerometer worn on 7 consecutive nights, presence of diagnosis of obstructive sleep apnea, and diffusion tensor imaging indices of white matter integrity in 29 non-demented adults with Down Syndrome (48% female, aged 33–54 years). Average total sleep time was associated with lower mean diffusivity in the left superior longitudinal fasciculus (*r* = −0.398, *p* = 0.040). Average sleep efficiency, length of awakenings, and movement index were related to fractional anisotropy in the right inferior longitudinal fasciculus (*r* = −0.614 to 0.387, *p* ≤ 0.050). Diagnosis of obstructive sleep apnea was associated with fractional anisotropy in the right inferior longitudinal fasciculus (*r* = −0.373, *p* = 0.050). Findings suggest that more disrupted sleep is associated with lower white matter integrity in the major association tracts in middle-aged adults with Down syndrome. Longitudinal work is needed to confirm the directionally of associations. Sleep interventions could be an important component for promoting optimal brain aging in the Down syndrome population.

## 1. Introduction

Down Syndrome (DS) is a developmental disability caused by the full or partial triplication of chromosome 21 and occurs in approximately 1 in 700 live births in the United States each year [1]. Individuals with DS experience an early age of onset and increased risk for aging-related health conditions, including cognitive declines and Alzheimer’s disease, beginning in middle adulthood [2]. Included in this accelerated aging is a high prevalence of disordered sleep [3,4,5]. Obstructive sleep apnea, characterized by repetitive episodes of obstruction of the upper airway during sleep, is estimated to occur in 78 to 90% of adults with DS [6,7,8,9]. Adults with DS have also been shown to evidence difficulties with sleep initiation, sleep maintenance, and early awakening [10,11,12]. Indeed, across studies, about three-fourths of adults with DS included in samples have been found to evidence poor sleep efficiency (<85%) and are reported to spend an average of 103 to 126 min awake after sleep onset per night [9,13]. While sleep has been shown to become more fragmented with age in the general population, the severity of disrupted sleep (e.g., reduced total time, increased wake after sleep onset, and reduced time in REM sleep) reported for adults with DS is markedly worse than that of comparison groups of adults without DS matched on biological sex, age, and body mass index [9].

Little is known about brain pathology associated with the high prevalence of disordered sleep in adults with DS. Within the general population, recent studies have found self-reported and objectively-measured sleep disruptions to be related to white matter (WM) impairments [14,15,16] and that this may serve as a pathway linking disordered sleep to risk of dementia and stroke [17,18]. For example, Yaffe and colleagues (2016) found that shorter sleep durations were associated with lower WM integrity in otherwise healthy middle-aged adults as assessed by diffusion tension imaging indices of fractional anisotropy, mean diffusivity and WM hyperintensities. Kocevska and colleagues (2019) similarly found that middle-aged and older adults from general population samples who evidenced more wake time after sleep onset, lower sleep efficiency, and shorter total sleep time, as assessed by an actigraph accelerometer, had worse microstructures of cerebral WM via diffusion tensor imaging with alterations most evident in the major association tracts (e.g., superior and inferior longitudinal fasciculus), cingulum, and anterior forceps of the corpus callosum. Studies have also found self-reported disrupted sleep (e.g., self-report of difficulties with initiating and maintaining sleep) to be associated with decreased global fractional anisotropy [15]. Obstructive sleep apnea has also been linked to WM degeneration within the middle-aged and older adult general population [19,20,21,22]. For example, Baril and colleagues (2021) found higher WM hyperintensities in middle-aged and older adults with moderate-to-severe obstructive sleep apnea versus those without sleep apnea. Chen and others (2015) similarly found that middle-aged and older adults with obstructive sleep apnea had lower fractional anisotropy across various regions, including the left superior longitudinal fasciculus versus those without obstructive sleep apnea. Potential mechanisms linking disrupted sleep and obstructive sleep apnea to WM degeneration have been posited to include inflammation, amyloid deposition, hemodynamic changes and oxidative stress, caused by intermittent hypoxemia and sleep fragmentation [23]. To date, little is known about the association between sleep disruptions or sleep disordered breathing and WM integrity in adults with DS. It is critical to understand these connections given the high prevalence of sleep problems in DS and the role of WM integrity for optimal aging in a population at risk for early age cognitive decline and Alzheimer’s disease [24] and other aging-related conditions [9,25,26].

The goal of the present study was to determine the association between disrupted sleep and WM integrity in a sample of 29 non-demented adults with DS (aged 33–54 years). Sleep was assessed using a wrist-worn actigraph accelerometer on 7 consecutive nights and caregiver-report of the presence of medical diagnosis of obstructive sleep apnea. WM integrity was assessed using fractional anisotropy and mean diffusivity via Diffusion Tensor Imaging in the association tracts (i.e., left and right superior and inferior longitudinal fasciculus). In line with non-DS populations, we hypothesized that adults with DS who evidence more disrupted sleep (e.g., lower total sleep time, higher wake after sleep onset and lower sleep efficiency) and/or reported a medical diagnosis of obstructive sleep apnea would evidence greater WM degeneration than those without disrupted sleep. The current study is exploratory in nature given the small sample size; however, results will yield important and novel information about the connection between disrupted sleep and WM in DS and lay the groundwork for future larger and longitudinal studies. 

## 2. Materials and Methods

### 2.1. Participants

Twenty-nine non-demented adults with DS (aged 33 to 54 years old) with confirmed full trisomy 21 participated in this study. Participants were part of a larger multi-site longitudinal study ABC-DS (Alzheimer’s Biomarkers Consortium—Down Syndrome) tracking biomarkers of Alzheimer’s disease that involved caregiver-report of health history, cognitive testing and neuroimaging (for full methods see Handen et al., 2021). Participants were recruited from two ABC-DS sites—University of Wisconsin-Madison and University of Pittsburgh. Study inclusion criteria were: being 25 years or older, having a mental age of 30 months or greater, genetic testing to confirm trisomy 21, no contradictive conditions for neuroimaging scans (i.e., pregnant or breastfeeding, metal in body, or claustrophobia), no conditions that impaired cognitive functioning, and no clinical diagnosis of Alzheimer’s disease at entry into the study. The study was approved by both sites Institutional Review Boards, by a properly constituted research ethics committee.

### 2.2. Procedure

As part of ABC-DS, the adult with DS and their caregiver completed a two-day study visit. On the first day, the adult with DS was administered a battery of neuropsychological tests while the caregiver completed surveys on the health and functioning of the DS participant. On the second day, the adult with DS completed MRI sequences including diffusion tensor imaging. After completing the visits, the individual with DS was sent home with a GT9x actigraph accelerometer and instructed to wear it on their non-dominant hand for 7 consecutive days and nights. The individual with DS and their caregiver completed a daily sleep record during the 7 days, which was used to validate the sleep actigraph data.

### 2.3. Measures

#### 2.3.1. Diffusion Tensor Imaging MRI

MRI acquisition was used for diffusion tensor imaging to examine microstructural differences in water diffusion properties of tissue [27]. The current study focused on mean diffusivity which is the directionally averaged diffusivity and index of the density of microstructural features. We also focused on fractional anisotropy, which is a normalized directional variance of diffusivities. The diffusion tensor imaging protocol consisted of 4 (University of Pittsburgh) or 6 (University of Wisconsin—Madison) non-diffusion weighted (b_0_) images and diffusion weighted images with a b-value of 1000 s/mm^2^ in 48 non-colinear directions, matrix size: 128 × 128 with 56 slices, field of view: 24 × 24 × 24 cm^3^ and 2.5 mm slice thickness. In-house processing from FSL [28] and the DiPy toolbox [29] were used. Diffusion weighted data were corrected for Gibbs’ ringing artifacts and eddy current distortions and head motion with outlier replacement. The diffusion tensors were estimated using RESTORE [30]. Regions of interest were the major tract associations, superior longitudinal fasciculus and inferior longitudinal fasciculus.

#### 2.3.2. Disrupted Sleep

The ActiGraph GT9X sampled movement data at a sample rate of 30 HZ and epoch length of 60 s. To identify sleep-wake times, we used the Cole–Kripke algorithm [31], a validated algorithm on ActiLife (Version 6.13.1). Variables included: total sleep time, wake after sleep onset, sleep efficiency, number of awakenings, average length of awakenings (in minutes), movement index, and sleep fragmentation index. The average (i.e., mean across the 7 nights) for each variable was used in analyses. Actigraphy was collected across 7 consecutive nights in order to capture both weekday and weekend sleep patterns, A 7 nights of actigraphy data has been found to have adequate validity with other objective measures of sleep [32].

Participants and caregivers also completed a daily sleep log for all 7 nights which included time in and out of bed, number of times woke up during the night, number of minutes the participant was awake after falling asleep, and questions about sleep quality. This subjective sleep log was used to validate actigraph data and ensure participant reported in and out of bed times were aligned with actigraphy timing of sleep. Studies have shown that sleep logs are useful in validating actigraph parameters [33,34].

#### 2.3.3. Obstructive Sleep Apnea

Caregiver-reported obstructive sleep apnea was also used to assess sleep within our population. Caregivers reports on the presence (or absence) of obstructive sleep apnea, were coded 1 = present and 0 = absent. Study staff assisted caregivers on the health form by answering questions and providing explanations, as needed.

### 2.4. Data Analysis

Descriptive statistics were first used to examine the distribution of study variables and identify any outliers. Pearson correlations were conducted to examine the association between socio-demographics and control variables (chronological age, biological sex, and level of intellectual disability) and the sleep and WM variables. Partial correlations were used to examine the association between sleep and WM integrity (fractional anisotropy and mean diffusivity in the association tracts) after partialing out relevant (i.e., those significantly associated with sleep or WM variables) with the superior and inferior longitudinal fasciculi. An alpha of 0.05 was used to determine statistical significance despite multiple comparisons given our small sample size.

## 3. Results

Within this sample of non-demented adults with DS (Table 1), the average age was 40.41 years (*SD* = 6.73), with about half of participants having moderate intellectual disability (n = 15, 52%). Approximately half of the participants were female (n = 14, 48%). Twenty-eight participants (97%) had valid actigraph data for 6 or more nights, while one participant had 5 nights of valid actigraph data (34%). Thus, number of valid nights of actigraph data (5, 6, or 7) was controlled for in remaining analyses using actigraph variables. Descriptive data indicated that the actigraph variables sleep efficiency and length of awakenings were skewed (kurtosis: 2.708 and 3.669, respectively). Given this skew, log transformations were conducted for these variables and these transformed scores were used in addition to raw scores. The pattern of significant associations did not differ when using transformed versus raw scores and those we report are with models using the raw scores below to aid in interpretability. All other variables were normally distributed (kurtosis: −0.503 to 2.001). 

On average, participants had a sleep efficiency of 77% (*SD* = 11.37), slept an average of 7.16 h per night (*SD* = 1.39), and woke up 21 times per night (*SD* = 11.37) for an average length of awakenings of 5 min (*SD* = 2.37). Fourteen (49%) participants were reported to have been given a medical diagnosis of obstructive sleep apnea. There was a significant association between sleep record reported variables of average time between in and out of bed time, total sleep time and actigraphy variables of total sleep time (*r* = 0.610 to 0.668, ps > 0.05). Additionally, sleep efficiency, WASO and sleep record reported feeling of being refreshed (*r* = −0.278 to 0.312, ps > 0.05). The presence of a medical diagnosis of obstructive sleep apnea was significantly associated with actigraphy variable of movement index (*r* = 0.380, *p* = 0.048) but was not significantly related to remaining actigraphy variables (*r* = −0.143–0.321, ps > 0.05).

Pearson correlations were conducted to examine the association between main study variables and socio-demographics (chronological age, biological sex, intellectual disability). Chronological age was associated with number of awakenings (*r* = −0.374, *p* = 0.046), fractional anisotropy in the superior longitudinal fasciculus left (*r* = −0.498, *p* = 0.006) and right (*r* = −0.510, *p* = 0.005), fractional anisotropy in the inferior longitudinal fasciculus left (*r* = −0.551, *p* = 0.002) and right (*r* = −0.599, *p* = 0.001), mean diffusivity in the superior longitudinal fasciculus left (*r* = 0.731, *p* = 0.001) and right (*r* = 0.651, *p* = 0.001), mean diffusivity in the inferior longitudinal fasciculus left (*r* = 0.636, *p* = 0.001) and right (*r* = 0.655, *p* = 0.001). There were no significant correlations between main study variables and intellectual disability or biological sex (rs = −0.276 to 0.476, and ps 0.148 to 0.961), so these were not controlled for in remaining analyses.

### WM Integrity and Sleep

After controlling for chronological age and number of valid nights of actigraph data, partial correlations indicated that more disrupted sleep was associated with lower WM integrity (Table 2). Specifically, lower total sleep time was significantly associated with higher mean diffusivity in the left superior longitudinal fasciculus (*r* = −0.394, *p* = 0.040). Sleep efficiency was significantly positively correlated with fractional anisotropy in the left inferior longitudinal fasciculus (*r* = 0.387, *p* = 0.046). Length of awakenings was significantly negatively associated with fractional anisotropy in the left (*r* = −0.425, *p* = 0.027) and right (*r* = −0.614, *p* = 0.001) inferior longitudinal fasciculus. The movement index was significantly negatively correlated to fractional anisotropy in the right inferior longitudinal fasciculus (*r* = −0.430, *p* = 0.025). After controlling for chronological age, partial correlations indicated that the presence of a medical diagnosis of sleep apnea was significantly negatively associated with fractional anisotropy in the right inferior longitudinal fasciculus (*r* = −0.373, *p* = 0.050). There were no other significant correlations.

Given the role of obesity in disordered sleep, follow-up analyses were run to examine the relation between sleep and WM integrity, controlling for body-mass-index. Total sleep time continued to be significantly negatively correlated with mean diffusivity in the left superior longitudinal fasciculus (*r* = −0.431, *p* = 0.031). Length of awakening continued to be significantly negatively correlated with fractional anisotropy in the left (*r* = −0.447, *p* = 0.025) and right (*r* = −0.602, *p* = 0.001) inferior longitudinal fasciculus. Movement index continued to be significantly negatively associated with the right inferior longitudinal fasciculus (*r* = −0.421, *p* = 0.036). The association between diagnosis of obstructive sleep apnea and lower fractional anisotropy in the right inferior longitudinal fasciculus remained at a trend-level (*r* = −0.370, *p* = 0.058). There were no additional significant correlations.

## 4. Discussion

Disrupted sleep has been estimated to occur in one-third or more of the adult DS population and up to three-fourths of adults with DS are estimated to have obstructed sleep apnea [6,12]. In non-DS populations, there is growing evidence that disrupted sleep (such as less total sleep time, worse sleep efficiency, and more wakes after sleep onset) and obstructive sleep apnea are related to poorer WM integrity [14,15,16]. The current study was among the first to explore the relation between disrupted sleep and obstructive sleep apnea and WM integrity in middle-aged adults with DS. Consistent with studies from the general population [15,20,22], and in support of our hypotheses, greater sleep disruptions (i.e., less total sleep time, worse sleep efficiency, longer lengths of time awake, and more movement during sleep) were related to lower fractional anisotropy and higher mean diffusivity in the inferior and superior longitudinal fasciculus, indicating lower WM integrity in these association tracts. In addition, lower integrity of WM in the right inferior longitudinal fasciculus, as assessed by lower fractional anisotropy in this region, was associated with the presence of diagnosed obstructive sleep apnea. These findings were in models controlling for chronological age and thus occur above and beyond any independent effects of aging on WM integrity. Moreover, in follow-up analyses, after controlling for BMI, the majority of the associations between disrupted sleep and WM integrity remained significant, and the association between diagnosis of obstructive sleep apnea and WM integrity was at a trend-level. Thus, disrupted sleep and sleep disordered breathing appear to have relations with WM integrity independent of body mass index.

Further longitudinal research is needed to understand the mechanisms linking disrupted sleep to WM impairment in DS. In the general population, evidence suggests that disrupted sleep and obstructive sleep apnea may accelerate the process of WM degeneration through several mechanisms including oxidative stress and/or inflammation [20,35,36]. These mechanisms may lead to WM loss as well as cellular shrinkage, and could play a role in accelerated aging, including Alzheimer’s disease [37,38,39,40], a condition of which adults with DS have a lifetime prevalence of 90% [41] and a condition that has been linked to disrupted sleep in middle-aged adults with DS [13,42]. Future longitudinal studies are needed to determine if disrupted sleep and obstructive sleep apnea are linked to WM loss through these potential mechanisms in adult with DS, and if so, whether this pathology contributes to variability in the timing of Alzheimer’s disease in the DS population. It is possible that intervention efforts that target reducing sleep disruptions and sleep disordered breathing problems could offer a way to maintain WM integrity in aging adults with DS, and that these efforts could also have beneficial effects on other aging-related conditions such Alzheimer’s disease. If such effects are borne out in longitudinal studies, behavioral interventions (e.g., psychoeducation for good sleep hygiene and body positioning to promote airway openings during the night) and medications aimed at reducing effects of sleep disruptions on REM cycles (e.g., cholinergic agonists) could be used. Biomedical interventions (e.g., the use of CPAP and BPAP) to reduce sleep disordered breathing are likely to play a critical role in such interventions. Developing strategies for increasing compliance with such devices will be important for the field given evidence that compliance in everyday life with such devices is often very low [43,44].

It is important to note that given the concurrent nature of this study, the current study is not able to determine if disrupted sleep and/or obstructive sleep apnea lead to WM loss. While there is support for this pathway, it is also possible that disrupted sleep results from WM loss and/or that both disruptive sleep and WM loss are driven by other mechanisms (e.g., cardiovascular health). It is possible that mental health conditions such as depressive disorders contribute to disrupted sleep in adults with DS and would need to be targeted as well. These questions are of high importance to the field and should be evaluated in future studies. The current study had both strengths and limitations. Strengths included the use of direct measures of disrupted sleep (actigraph accelerometer) and WM integrity (MRI-based diffusion tensor imaging). Limitations of the study included the cross-sectional, exploratory design, which prohibits understanding of the directionality of the association between sleep and WM integrity. The study is also limited in terms of measuring obstructive sleep apnea via caregiver-report of medical diagnosis as opposed to objective measures of disordered breathing during sleep. The use of sleep apnea as a dichotomous variable did not allow for severity of the disorder (e.g., apnea–hypopnea index, hypoxic burden, etc.) to be considered in statistical analyses. The prevalence of obstructive sleep apnea (49%) in our sample is on the low end of what has been reported by others^6^, suggesting that some of our participants could have undiagnosed obstructive sleep apnea. Indeed, many caregivers reported that the adult with DS had not been tested for sleep disordered breathing problems. In future studies it will also be important to consider whether or not sleep disordered breathing is being treated. Lastly, the study sample was small, and findings are exploratory given that significance was set at an alpha of 0.05 despite multiple analyses and the examination of multiple brain association tracks. Future studies with include larger samples are needed to replicate findings.

## 5. Conclusions

In summary, disrupted sleep (i.e., less sleep time, lower sleep efficiency, longer awakenings, and more movement at night) was associated with lower WM integrity in the superior longitudinal fasciculus and inferior longitudinal fasciculus association tracts in the current sample of middle-aged non-demented adults with DS in models controlling for chronological age. A diagnosis of obstructive sleep apnea was associated with lower WM integrity in the right inferior longitudinal fasciculus. If findings are borne out in larger longitudinal studies, they may suggest that sleep is a modifiable factor that could be targeted in intervention as a way to maintain WM integrity in DS. This may have beneficial effects for promoting healthy brain aging, including possibly delaying the onset of Alzheimer’s disease, as both disrupted sleep [13] and WM impairment [42] have been implicated in Alzheimer’s disease in DS. 

## Figures and Tables

**Table 1 brainsci-11-01322-t001:** Participant characteristics and mean and Standard Deviation for the study variables.

	Total (N = 29)
Female, No. (%)	14 (48.3)
Chronological age in years, M (SD)	40.41 (6.73)
Intellectual disability level, No. (%)	
Mild	14 (48.3)
Moderate	15 (51.7)
Body mass index ^e^, M (SD)	34.45 (9.06)
White matter integrity, M (SD)	
Superior longitudinal fasciculus, left FA	0.43 (0.03)
Superior longitudinal fasciculus, right FA	0.45 (0.03)
Inferior longitudinal fasciculus left, FA	0.48 (0.03)
Inferior longitudinal fasciculus, right FA	0.50 (0.04)
Superior longitudinal fasciculus, left MD	0.74 (0.04)
Superior longitudinal fasciculus, right MD	0.74 (0.03)
Inferior longitudinal fasciculus, left MD	0.85 (0.05)
Inferior longitudinal fasciculus, right MD	0.82 (0.06)
Sleep	
Total sleep time, M (SD)	429.87 (83.42)
Wake after sleep onset, M (SD)	107.96 (59.68)
Sleep efficiency M (SD)	76.85 (11.37)
Number of awakenings, M (SD)	21.61 (11.37)
Length of awakenings (min), M (SD)	4.90 (2.37)
Movement index, M (SD)	26.18 (10.68)
Sleep fragmentation index M (SD)	43.16 (16.52)

Note: Unless otherwise indicated, data are expressed as mean (SD). Intellectual disability levels reflect the following mental age: mild: ≥9 years, moderate: 4 to 8 years. Sleep variables are average over a 7-day period. FA = fractional anisotropy. MD = mean diffusivity (in 10^−3^ mm^2^/s). ^e^ Calculated as weight in kilograms divided by height in meters squared. Percent daytime in sedentary and moderate-to-vigorous calculated with sleep as non-wear time.

**Table 2 brainsci-11-01322-t002:** Associations between white matter integrity and actigraph sleep indices and diagnosis of obstructive sleep apnea (n = 29).

		TST ^1,2^	WASO ^1,2^	SE ^1,2^	NOA ^1,2^	LOA ^1,2^	MI ^1,2^	SFI ^1,2^	OSA ^1^
Superior longitudinal fasciculus, left FA	Corr. Sig.	0.233 0.263	−0.240 0.227	0.323 0.100	−0.356 0.069	−0.206 0.302	−0.307 0.119	−0.284 0.152	−0.151 0.444
Superior longitudinal fasciculus, right FA	Corr. Sig.	0.135 0.503	−0.220 0.270	0.279 0.158	−0.238 0.232	−0.264 0.183	−0.291 0.140	−0.193 0.334	−0.190 0.332
Inferior longitudinal fasciculus left, FA	Corr. Sig.	0.295 0.136	−0.204 0.308	0.328 0.095	0.027 0.892	**−0.425** **0.027**	−0.317 0.107	−0.210 0.292	−0.229 0.242
Inferior longitudinal fasciculus, right FA	Corr. Sig.	0.279 0.159	−0.280 0.157	**0.387** **0.046**	0.081 0.689	**−0.614** **0.001**	**−0.430** **0.025**	−0.323 0.100	**−0.373** **0.050**
Superior longitudinal fasciculus, left MD	Corr. Sig.	**−0.398** **0.040**	0.131 0.514	−0.216 0.279	0.222 0.267	0.053 0.794	0.219 0.273	0.203 0.311	−0.111 0.574
Superior longitudinal fasciculus, right MD	Corr. Sig.	−0.201 0.315	0.007 0.972	−0.092 0.647	0.003 0.988	0.124 0.538	0.130 0.519	0.064 0.752	0.213 0.277
Inferior longitudinal fasciculus, left MD	Corr. Sig.	−0.025 0.901	−0.243 0.222	0.224 0.261	−0.218 0.274	−0.196 0.326	−0.288 0.145	−0.313 0.112	−0.054 0.785
Inferior longitudinal fasciculus, right MD	Corr. Sig.	−0.195 0.330	−0.092 0.646	−0.021 0.916	−0.255 0.199	0.167 0.404	0.096 0.633	0.034 0.867	0.266 0.172

Note. TST = total sleep time; WASO = wake after sleep onset; SE = sleep efficiency; NOA = number of awakenings; LOA = length of awakenings; MI = movement index; SFI = sleep fragmentation index; OSA = obstructive sleep apnea; FA = fractional anisotropy; MD = mean diffusivity; Sig. = Significance (2-tailed); Corr. = Correlations. **Bolded text** = *p* < 0.05. ^1^ = controlling for chronological age; ^2^ = controlling for number of valid nights of actigraphy data.

## Data Availability

The data that support the findings of this study are available from the corresponding author upon request.

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
