# Peer review of "Sleep and White Matter in Adults with Down Syndrome"

_brainsci, 2021, doi:10.3390/brainsci11101322_

Round 1

Reviewer 1 Report

The current manuscript provides the association between disrupted sleep and white matter integrity in a sample of 29 non-demented adults with Down’s syndrome (DS) by hypothesizing that adults with DS who evidence more disrupted sleep (e.g., lower total sleep time, higher wake 84 after sleep onset and lower sleep efficiency) and/or reported a medical diagnosis of obstructive sleep apnea would evidence greater WM degeneration than those without disrupted sleep. This is well written manuscript, however following minor points needs to be addressed.

  1. Reason for choosing only 7 days of sleep cycle for actigraphy data record.
  2. How about other conditions like depression affects the sleeping pattern in DS individuals?
  3. Overall measures to be taken to improve the sleep efficiency needs to be included in the discussion.

Author Response

Reason for choosing only 7 days of sleep cycle for actigraphy data record.

RESPONSE: We have added (page 3) clarification that actigraphy was collected across 7 consecutive nights in order to capture both weekday and weekend sleep patterns. Moreover, 7 nights of actigraphy data has previously been found to have adequate validity with other objective measures of sleep (Ali et al., 2017).

Aili, K., Åström-Paulsson, S., Stoetzer, U., Svartengren, M., & Hillert, L. (2017). Reliability of Actigraphy and Subjective Sleep Measurements in Adults: The Design of Sleep Assessments. Journal of clinical sleep medicine : JCSM : official publication of the American Academy of Sleep Medicine13(1), 39–47. https://doi.org/10.5664/jcsm.6384

How about other conditions like depression affects the sleeping pattern in DS individuals?

RESPONSE: This is a good point. Mental (e.g., depression) and physical health (e.g., cardiovascular) conditions that are elevated in the Down syndrome population could be driving some of the associations we found between sleep and white matter impairments. This possibility and the need for future research is now highlighted on page 7.

Overall measures to be taken to improve the sleep efficiency needs to be included in the discussion

RESPONSE: In accordance with this recommendation, we have expanded on our discussion of behavioral and biomedical interventions that may be leveraged to improve sleep efficiency in Down syndrome (page 7).

Reviewer 2 Report

This report from Fleming et al., entitled Sleep and White Matter in Adults with Down Syndrome, describes the results of a small study investigating the relationship between poor sleep quality and white matter integrity in a small cohort of patients affected by Down Syndrome. While similar studies have been reported in the past, this report suggests interesting opportunities to favour healthy aging in patients with DS. In its current form, the manuscript presents a good amount of interesting findings that are supported by the reported data. Thus overall I recommend the publication of this manuscript in Brain Sciences after minor modifications.

The results are well presented and sufficient to draw relevant conclusions. The analyses are appropriate and clear. The authors readily identify the important limitations of the study, mainly the small number of individuals constituting this cohort.  

Minor points:

The authors claim the data recorded from the accelerometer were validated using a sleep record, the authors should add more details about this process and indicate the parameters followed for this validation.

The authors recognise that some participant might have undiagnosed OSA, it would be interesting to verify whether OSA diagnosis was associated with greater sleep disruptions.

Table1, BMI does not show SD.

The hypothesis of targeting sleep quality to maintain WM integrity is intriguing, the authors should discuss further this point, focusing on the relevance and the advantages for DS patients.

Suggest revising the text, line 248

Author Response

The authors claim the data recorded from the accelerometer were validated using a sleep record, the authors should add more details about this process and indicate the parameters followed for this validation.

RESPONSE: This is a great point. We have added information regarding the association between the sleep records of in and out of bed times and the acigraphy variables on page 4.

The authors recognize that some participant might have undiagnosed OSA, it would be interesting to verify whether OSA diagnosis was associated with greater sleep disruptions.

RESPONSE: On page 4, we have added a statement regarding the association between diagnosed OSA and actigraphy variables.

Table1, BMI does not show SD.

RESPONSE: We have added the SD for BMI into table 1. Thank you for pointing this out.

The hypothesis of targeting sleep quality to maintain WM integrity is intriguing, the authors should discuss further this point, focusing on the relevance and the advantages for DS patients.

RESPONSE: In response to this recommendation, we have elaborated on our discussion of the potential relevance of targeting sleep for maintaining WM integrity and healthy aging in Down syndrome given its unique aging and Alzheimer’s disease profile (Page 7).

Suggest revising the text, line 248

RESPONSE: We have revised line 248 to (Page 7).